# Dry heat sterilization as a method to recycle N95 respirator masks: The importance of fit

John G. Yuen[1][☯], Amy C. Marshilok[2,3,4][☯], Peter Todd Benziger[5,6], Shan Yan[2,3], Jeronimo Cello[5,6], Chavis A. Stackhouse[3,7], Kim Kisslinger[8], David C. Bock[2,3], Esther S. Takeuchi[2,3,4], Kenneth J. Takeuchi[2,3,4], Lei Wang[2,3], Sruthi Babu[1], Glen Itzkowitz[9], David Thanassi[5,6], Daniel A. Knopf[10]*, Kenneth R. Shroyer[1]*

1 Department of Pathology, Renaissance School of Medicine, Stony Brook University, Stony Brook, New York, United States of America, 2 Interdisciplinary Science Department, Brookhaven National Laboratory, Upton, New York, United States of America, 3 Institute for Electrochemically Stored Energy, Stony Brook University, Stony Brook, New York, United States of America, 4 Department of Materials Science and Chemical Engineering, Stony Brook University, Stony Brook, New York, United States of America, 5 Department of Microbiology and Immunology, Stony Brook University, Stony Brook, New York, United States of America, 6 Center for Infectious Diseases, Stony Brook University, Stony Brook, New York, United States of America, 7 Department of Chemistry, Stony Brook University, Stony Brook, New York, United States of America, 8 Center for Functional Nanomaterials, Brookhaven National Laboratory, Upton, New York, United States of America, 9 Office of the Dean, Renaissance School of Medicine, Stony Brook University, Stony Brook, New York, United States of America, 10 School of Marine and Atmospheric Sciences, Stony Brook University, Stony Brook, New York, United States of America

☯ These authors contributed equally to this work.
* Daniel.Knopf@stonybrook.edu (DAK); Kenneth.Shroyer@stonybrookmedicine.edu (KRS)

**Data Availability Statement:** All relevant data are within the manuscript and its Supporting information files.

## Abstract

In times of crisis, including the current COVID-19 pandemic, the supply chain of filtering facepiece respirators, such as N95 respirators, are disrupted. To combat shortages of N95 respirators, many institutions were forced to decontaminate and reuse respirators. While several reports have evaluated the impact on filtration as a measurement of preservation of respirator function after decontamination, the equally important fact of maintaining proper fit to the users' face has been understudied. In the current study, we demonstrate the complete inactivation of SARS-CoV-2 and preservation of fit test performance of N95 respirators following treatment with dry heat. We apply scanning electron microscopy with energy dispersive X-ray spectroscopy (SEM/EDS), X-ray diffraction (XRD) measurements, Raman spectroscopy, and contact angle measurements to analyze filter material changes as a consequence of different decontamination treatments. We further compared the integrity of the respirator after autoclaving versus dry heat treatment via quantitative fit testing and found that autoclaving, but not dry heat, causes the fit of the respirator onto the users face to fail, thereby rendering the decontaminated respirator unusable. Our findings highlight the importance to account for both efficacy of disinfection and mask fit when reprocessing respirators to for clinical redeployment.

## Introduction

The transmission of SARS-CoV-2, the etiologic agent of COVID-19, is predominantly by aerosol, therefore N95 respirators, which are intended to exclude 95 percent of particulates in the

**Funding:** Contact angle, Raman spectroscopy, and electron microscopy characterization were supported by the DOE Office of Science through the National Virtual Biotechnology Laboratory, a consortium of DOE national laboratories focused on response to COVID-19, with funding provided by the Coronavirus CARES Act (https://science.osti.gov/nvbl). Electron microscopy data were collected at Center for Functional Nanomaterials, which is a U.S. DOE Office of Science Facility, at Brookhaven National Laboratory under Contract No. DE-SC0012704 (https://www.bnl.gov). C.A.S. acknowledges support from the NIH Institutional Research and Academic Career Development Award and New York Consortium for the Advancement of Postdoctoral Scholars (IRACDA-NYCAPS), award K12-GM102778 (https://www.nigms.nih.gov).

**Competing interests:** The authors have declared that no competing interests exist.

size range that encompasses most aerosolized viral droplets, including SARS-CoV-2, are recommended for protection of health care providers during patient encounters [1,2]. During the COVID-19 pandemic, the supply chain of personal protective equipment (PPE) for healthcare workers was pushed to its limit [3,4], necessitating the implementation of various protocols for the reuse of PPE by healthcare facilities throughout the world (10). The Centers of Disease Control and Prevention (CDC) in the United States recently issued additional guidance for the reuse of filtering facepiece respirators (FFR), such as N95 respirators, when there are shortages of respirators at healthcare facilities [5].

While fomite transmission of SARS-CoV-2 is unlikely to be a major source of virus transmission in the general population, minimizing its risk in healthcare workers is still an important consideration [6]. SARS-CoV-2 surface stability is affected by multiple factors, including the material that it contacts, the relative humidity of the environment, and the temperature at which it is exposed to. During times of crisis, both the CDC and other organizations including 3M, a major respirator manufacturer, have frequently cited ultraviolet germicidal irradiation, vaporous hydrogen peroxide, and moist heat as recommended methods for decontamination of FFRs [7–10]. These methods, however, often call for specific equipment that may prove difficult to obtain and/or difficult to implement in many healthcare facilities and in the general public.

Heat is potentially more readily accessible than other methods of decontamination in many healthcare facilities. Although autoclaving (i.e., steam at ~121˚C and > 15 psi) is a proven method of sterilization of most pathogens, it is not viable for decontamination of used N95 respirators because moist heat can degrade filter efficiency [11]. In culture medium, SARS-CoV-2 has been reported to be inactivated by dry heat treatment in as little as 5 minutes at 70˚C [12] and exposure of SARS Cov-2 on surfaces to dry heat at >70˚C for >30 minutes is sufficient to achieve a ≥3-log reduction of viral titers, meeting FDA recommendations for FFR reuse [13–15].

While several reports have evaluated the impact on filtration as a measurement of preservation of respirator function after decontamination [16], the equally important fact of maintaining proper fit to the wearer's face has been understudied. Per both the National Institute for Occupational Safety and Health Part 84 Title 42 of the Code of Federal Regulations (NIOSH 42 CFR 84) and the FDA, respirators must be assessed for not only filter performance, but also fit, i.e., the sealant performance between mask on the individual's face. Respirator mask fit testers such as the TSI PortaCount Pro 8048 are employed to rapidly obtain Occupational Health and Safety (OSHA)-compliant fit factors that quantitatively evaluate whether a respirator fits properly on an individual's face. The fit factor is derived by comparing particle counts outside the respirator with ones inside the respirator. Clearly, the respirator fit test only passes if filtration material and seal to face are both in order. In other words, if the fit test is successful, it implicitly means that also the filtration material is operating satisfactorily. If the fit test fails (fit factor < 100), it can either mean that the mask-to-face seal or filtration material failed [17]. Thus, a successful fit test implies that a given decontamination method is not altering the filtration material and respirator fit significantly.

In the current study, we demonstrate the complete inactivation of SARS-CoV-2 and preservation of fit test performance of N95 respirators following treatment with dry heat. We apply scanning electron microscopy with energy dispersive X-ray spectroscopy (SEM/EDS), X-ray diffraction (XRD) measurements, Raman spectroscopy, and contact angle measurements to analyze filter material changes as a consequence of different decontamination treatments. We further compared the integrity of the respirator after autoclaving versus dry heat treatment via quantitative fit testing and found that autoclaving, but not dry heat, causes the fit of the respirator onto the users face to fail, thereby rendering the decontaminated respirator unusable.

Our findings highlight the importance to account for both efficacy of disinfection and mask fit when reprocessing respirators to for clinical redeployment.

## Materials and methods

### N95 respirators

The class N95 filtering facepiece respirators chosen for this study include respirators typically used in large health care facilities:

- 3M™ Health Care Particulate Respirator and Surgical Mask 1860

- 3M™ Aura™ Health Care Particulate Respirator and Surgical Mask 1870+

- Bacou Willson 801 Respirator

- BLS 120B FFP1

### Heat treatment

N95 respirators were placed in a paper bag and sealed with a piece of heat-stable tape (Fig 1). The bags were placed onto a metal rack which was loaded into a TPS Gruenberg truck-in oven. Unless otherwise specified, all N95 respirators were treated for four cycles at either 80˚C for 60 minutes or 100˚C for 30 minutes, with at least 10 minutes of cooling time to room temperature in between. Autoclaved respirators were subjected to a single cycle of 121˚C at 15–25 psi for 30 minutes.

### SARS-CoV-2 thermal stability/virus recovery and quantification

The SARS-CoV-2 isolate USA-WA1/2020 was obtained from BEI Resources and used for the experiments in this study. VeroE6 cells were obtained from ATCC and used to titer and passage the SARS-COV-2 virus. VeroE6 cells were routinely cultured in DMEM containing

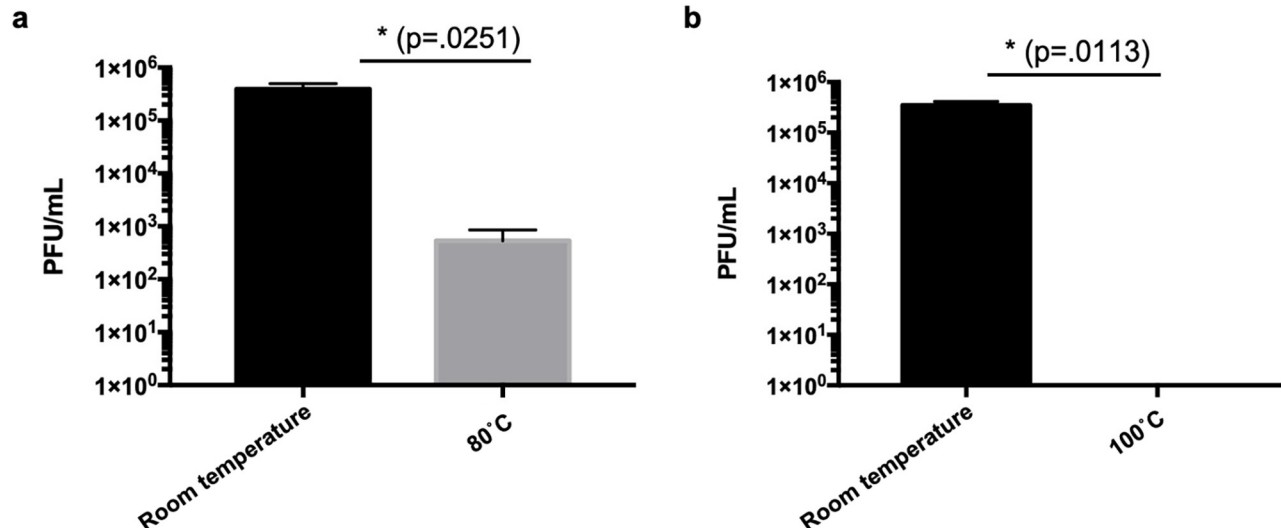

**Fig 1. SARS-CoV-2 thermal stability on N95 respirator material.** SARS-CoV-2 was inoculated onto N95 respirators and were subsequently subjected to either 80˚C of dry heat for 60 minutes or 100˚C of dry heat for 30 minutes.

Glutagro (Corning) and 8% Fetal Bovine Serum (FBS) at 37ºC with 5% $CO_2$. All growth and manipulations of the SARS-CoV-2 virus were performed under BSL3 containment conditions.

Approximately $5 \times 10^5$ PFU of SARS-CoV-2 virus in Dulbecco's Modified Eagle Medium (DMEM) was spotted in triplicate onto a N95 respirator for each condition and the respirators were left to dry within the biosafety cabinet for 2 hours at room temperature. N95 respirators containing SARS-COV-2 virus were either left at room temperature or treated with dry-heat using a TPS/Tenney T2 series (Tenney Environmental) small dry-heat sterilizer for the indicated time and temperature. As a negative control, N95 respirators were treated similarly with DMEM media alone and left at room temperature during heat treatment.

SARS-CoV-2 virus was recovered by cutting each virus-treated spot, including all three respirator layers, from each N95 respirator and placing each spot in an Eppendorf screw-cap tube containing 1 mL DMEM plus 10 units/ml penicillin, 10 µg/ml streptomycin, and 1 µg/ml amphotericin B. Samples were submerged and incubated for 5 minutes at room temperature then rocked gently by hand for 5 minutes to recover virus. Plaque assays were performed to quantify the amount of virus recovered by performing serial dilutions of recovered virus and infecting VeroE6 cells seeded at $4.5 \times 10^5$ cells/well in a 6-well tissue culture treated plate for one hour. Cells were then overlaid with DMEM containing 0.8% tragacanth gum, 2.5% FBS, 10 units/ml penicillin, 10 µg/ml streptomycin, and 1 µg/ml amphotericin B and incubated at 37ºC with 5% $CO_2$ for 48 hours. To quantify the amount of virus recovered, the overlay was removed, and the plaques were visualized by staining VeroE6 cells with 0.5% crystal violet and 0.8% glutaraldehyde in 50% methanol for 10 minutes followed by several washes with distilled water. Total PFU/mL for each condition was calculated by averaging the mean PFU/mL recovered for each biological replicate (n = 3).

## N95 FFR quantitative fit tests

A PortaCount Pro 8048 (TSI, Shoreview, MN) was used to obtain OSHA compliant quantitative fit factors at Stony Brook University Hospital, in the Department of Environmental Health and Safety [17]. The most penetrating particle size (MPPS) for most N95 respirators is around 300 nm [18–22]. For this reason, the quantitative fit testing protocols evaluate leakages at particle size ranges outside of the MPPS to be more sensitive to leakages across the sealant between respirator and face. The principle of operation is to choose a particle size, typically about 40 nm, that is filtered with great efficiency due to electrostatic interactions and to compare the number concentration of those particles outside the respirator (i.e. ambient air), the particle concentration inside the respirator [23,24]. Detection of those particles is achieved by a condensation particle counters (CPC) that grows these small particles via condensation of vapor to size detectable via light scattering. Quantitative fit testing was performed on the same operator for all respirator types and for all conditions: dry heat treated (n = 3), untreated (n = 1), and autoclaved (n = 1) respirators.

Quantitative fit testing procedures for N95 respirators were performed according to Occupational Safety and Health Administration (OSHA) guidelines found in Appendix A to §1910.134 [17]. N95 respirators were fitted with a respirator sampling adapter that allows for the measurement of particles inside the respirator while donned by an individual. Four exercises were performed (Table 1) and fit factors for each exercise were calculated by taking the ratio of the concentration of ambient particles to the concentration of particles inside the respirator. The overall fit factor is calculated as the ratio of the number of exercises (n) to the sum of the reciprocal of the fit factors for each exercise. Overall fit factor scores of $\geq 100$ passes

**Table 1. Description of OSHA guidelines on quantitative respirator fit testing procedures [17].**

| Exercises | Exercise Procedure | Measurement Procedure |
|---|---|---|
| 1) Bending Over | Bend at the waist, as if going to touch their toes for and inhale 2 times at the bottom. | 20 second ambient sample, followed by a 30 second respirator sample. |
| 2) Talking | The test subject will recite the Rainbow Passage loud enough to be heard by another person in the room. | 30 second respirator sample. |
| 3) Head Side-to-Side | Turn head from side to side for and inhale 2 times at each extreme. | 30 second respirator sample. |
| 4) Head Up-and-Down | Slowly move head up and down for inhale 2 times at each extreme. | 30 second respirator sample, followed by a 9 second ambient sample. |

OSHA guidelines.

$$Overall\ Fit\ Factor = \frac{n}{\sum_{k=1}^{n} \frac{1}{FitFactor_k}}$$

If a TSI Portacount Pro 8048 instrument is not available for respirator fit testing, in the supplement we have outlined procedures on how a scanning mobility particle sizer spectrometer (SMPS) consisting of a differential mobility analyzer (DMA) and condensation particle counter (CPC) could be used to estimate fit factors (S1 File).

## Filtration material characterization

Representative samples of respirator materials were obtained by cutting portions of untreated, dry air heat treated, and steam heat treated (autoclaved) respirators. Prior to SEM, Raman, and XRD measurements, both the Bacou Willson 801 N95 and 3M 1860 N95 FFR material samples were mechanically separated into three layers using forceps, where layer 1 is the outside layer (farthest from the mask wear), layer 2 is the middle meltblown layer and layer 3 is the inside layer (closest to the mask wearer).

*Scanning electron microscopy (SEM)* images were collected using a high-resolution SEM (JEOL 7600F) instrument. SEM images were acquired at an accelerating voltage of 5 kV. A thin layer of silver (Ag) 10 nm in thickness was applied to the respirator materials prior to SEM imaging to reduce sample charging.

Raman.

*Raman spectra* of the pristine and treated respirator materials were recorded on a Horiba Scientific XploRA instrument with a 532 nm laser at 10% intensity using a 50X objective and a grating of 1,200 lines/mm. The spectra were calibrated with a Si standard.

An artificial saliva (AS) solution was prepared with 0.844 mg/L NaCl, 1.200 mg/L KCl, 0.146 mg/L anhydrous $CaCl_2$, 0.052 mg/L $MgCl_2 \cdot 6H20$, 0.342 mg/L $K_2HPO_4$, 60.00 mg/L 70% sorbitol solution, 3.5 mg/L hydroxyethyl cellulose in deionized water.

**Contact angle measurements.** The contact angle as function of time was determined by use of Kyowa DM-501 instrument and measured with half angle method. Each experiment was run for 10 duplicate trials, using 20 μL artificial saliva solution, and data points were recorded every 100 milliseconds for 10 minutes, and the volume change as function of time was determined using the droplet profile and Kyowa FAMAS software.

## Statistical analysis

Viral titers were compared by calculating two-tailed *P* values using a Paired t test. Statistical analysis was performed using Prism 8 (GraphPad Software).

## Results

### SARS-CoV-2 thermal Stability on N95 respirators

SARS-CoV-2 thermal stability on 3M Particulate Respirator 1860 N95 material was evaluated by spotting $3 \times 10^5$ PFU of SARS-CoV-2 onto N95 respirators. After incubating the N95 respirators at 80˚C for 60 minutes, $1 \times 10^3$ PFU of viable virus was recovered from the respirator, demonstrating a 2-log reduction of virus as compared to samples that were kept at room temperature. Treatment of the inoculated N95 respirators at 100˚C however, returned no viable virus, demonstrating a $\geq$ 5-log reduction of virus (Fig 1).

### Quantitative fit testing of N95 respirators

Quantitative fit testing was performed on four models of N95 respirators after autoclaving or dry heat incubation at either 100˚C for 30 minutes or 80˚C for 60 minutes. For all respirator types, autoclaving resulted in failed quantitative fit testing (fit factor $<$ 100) (Fig 2). In contrast, dry heat incubation yielded passing fit test scores ($\geq$100) for all the respirators that passed quantitative fit testing prior to any treatment. None of the Bacou Willson 801 respirators passed quantitative fit testing, presumably due to poor fit on the user. It is notable, however, that respirators which were autoclaved yielded a lower fit factor than either the untreated, 100˚C, or the 80˚C dry heat treatment groups (Fig 2c).

### Material characterization of N95 respirators

The mesoscale morphologies of 3M 1860 N95 FFR were characterized by SEM before and after heat and autoclave treatment. The cross-section view was taken and application of EDS indicated that only carbon signal was detected at layers-1 and -2, while both carbon and oxygen were detected at layer-3 (Fig 3a and 3b). Layer-1 is ~300 μm in thickness, with millimeter scale patterning, comprised of microfibers with a diameter ~20 μm (Fig 3c). Layer-2 is ~300 μm in thickness, comprised of microfibers with a diameter in the range of 1–10 μm (Fig 3d). Layer-3 is ~1 mm in thickness, comprised of microfibers with a diameter ~30 μm, and some defects were also observed (Fig 3e). The morphologies of the 100˚C dry heat treatment (Fig 3f–3h) and autoclave treatment (Fig 3i–3k) of 3M 1860 N95 did not show obvious differences from SEM images.

Raman spectra were obtained for all of the layers of the 3M 1860 N95 FFR The broad asymmetric band observed at approximately 830 cm$^{-1}$ apparently splits into two bands at 808 and 840 cm$^{-1}$ upon crystallization. This indicates that the 830 cm$^{-1}$ band is a fundamental frequency of the chemical repeat unit that is altered by the symmetry of the helical chain conformation due to inter-molecular coupling between adjacent groups [25,26]. The 810 cm$^{-1}$ band can be assigned to helical chains within crystals, while a broader band at 840 cm$^{-1}$ assigned to chains in non-helical conformation [27].

Layer 1 showed significant fluorescence, as indicated by the broad peak features, possibly due to the dye used in this layer. Despite the strong fluorescence, a decrease in the ratio of two bands at 810 and 840 cm$^{-1}$ as well as the peak intensity at 972 cm$^{-1}$ after dry heat and autoclave treatment was noted, suggesting the shorting of the helical chain conformation of polypropylene after heat treatment (Fig 4a) [28]. Notably, the peak at 1220 cm$^{-1}$ ascribed to the helical chain of 14 monomeric units of polypropylene suggested shorting of the helical chain length. Layer 2 (middle layer) and layer 3 (inner layer) (Fig 4b and 4c) also contain polypropylene fibers that are lower in crystallinity and narrower in thickness. X-ray diffraction (XRD) analysis was also performed to identify the composition and crystallinity of each layer before and

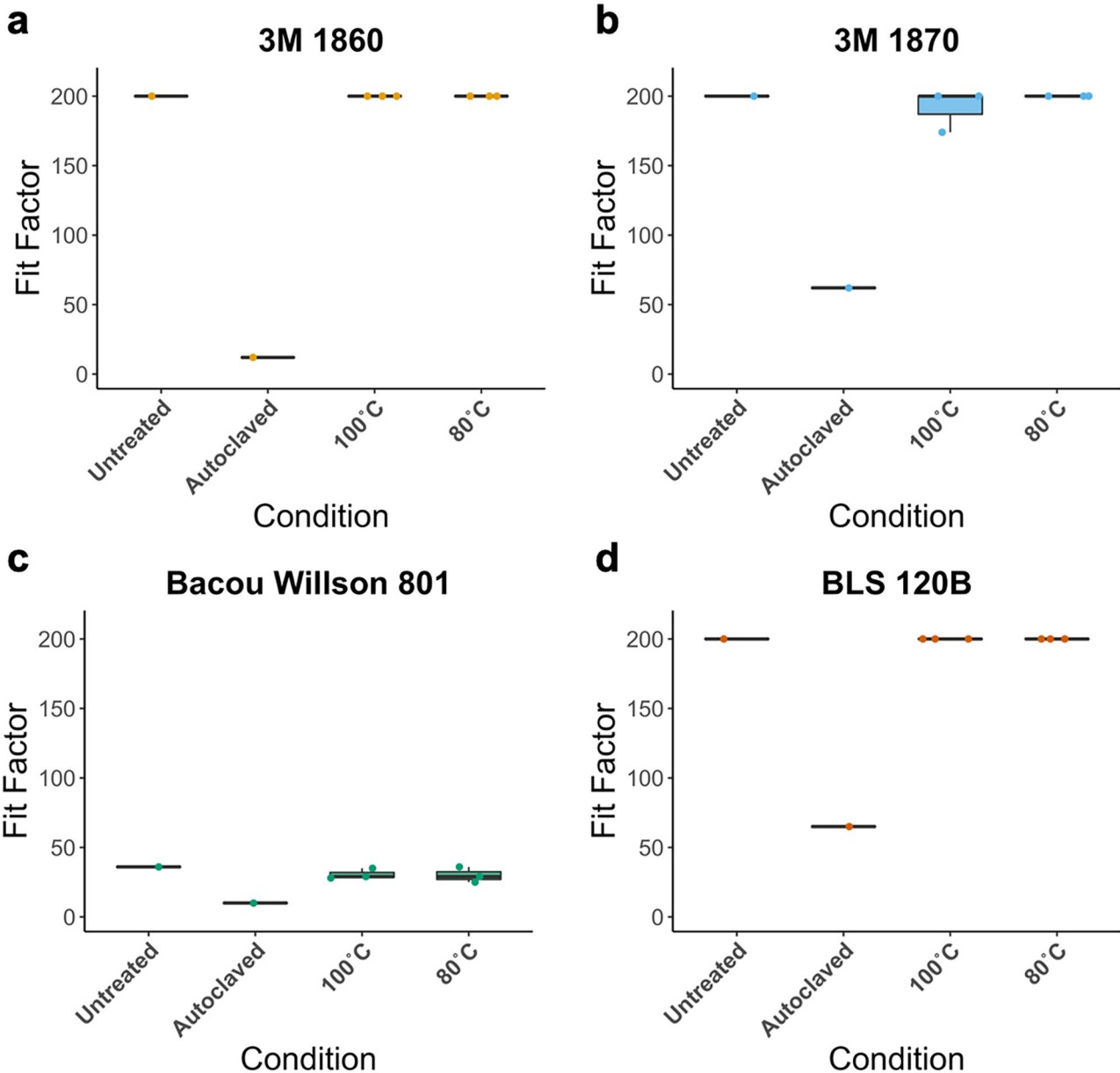

**Fig 2. Quantitative fit factors of N95 respirators.** Quantitative fit factors of N95 (**a**) 3M 1860, (**b**) 3M 1870, (**c**) Bacou Willson 801, and (**d**) BLS 120B respirators, treated with dry heat at 100˚C for 30' or 80˚C for 60' (n = 3), compared to untreated and autoclaved controls (n = 1).

after dry heat and autoclave treatment (S1 Fig), showing no compositional changes and insignificant crystallite sizes changes after both types of treatment.

Contact angle measurements were performed to characterize the wetting properties of surfaces of the respirator materials towards artificial saliva solution. Wetting describes the ability of a liquid to remain in contact with a given surface, and its qualities are dominated by van der Waals forces [29]. Contact angle data serves to indicate the degree of wetting when a liquid interacts with a solid. A contact angle greater than 90˚ suggests low wettability and poor contact of the fluid with the surface, resulting in a compact liquid droplet. Favorable wettability of surface evinces a contact angle less than 90˚ and the fluids will spread over a large area of the

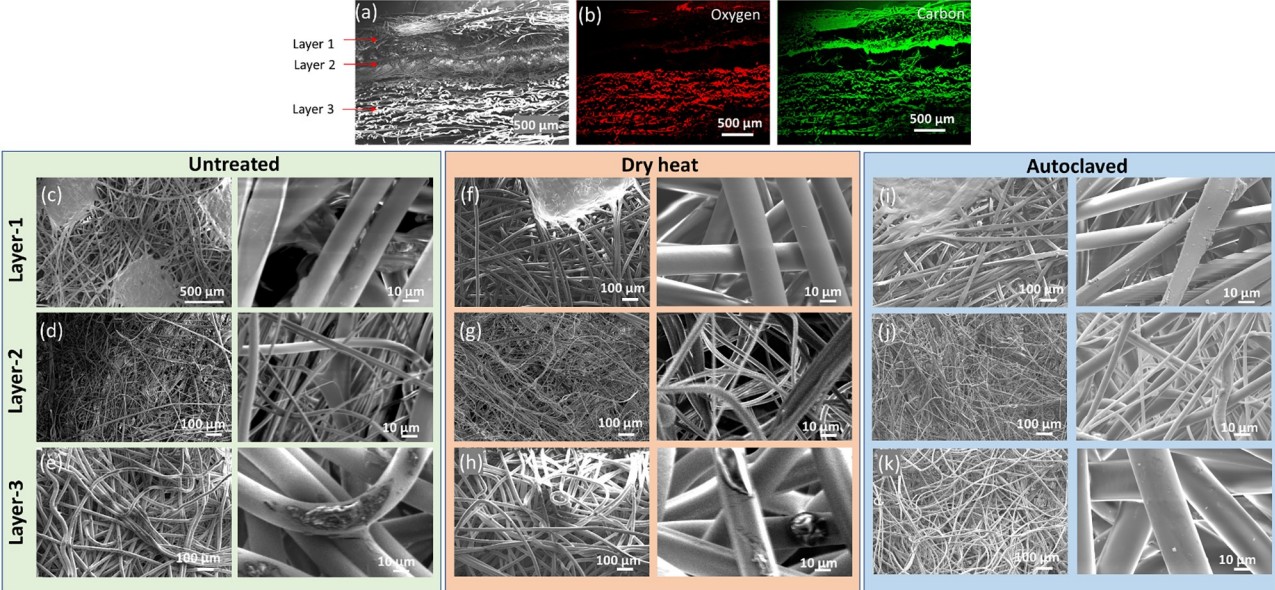

**Fig 3. SEM characterizations of three layers in 3M 1860 N95 FFR.** (**a-b**) SEM/EDS images of cross section. (**c-e**) SEM images of top-down view of untreated 3M 1860 N95. (**f-h**) SEM images of top-down view of the of 3M 1860 N95 after dry heat treatment at 100˚C for 4 cycles. (**i-k**) SEM images of top-down view of the of 3M 1860 N95 after autoclaving treatment.

measured surface. Saliva substitutes have been studied and are used in lieu of biological samples [30].

For the 3M 1860 N95 material, both the dry heat and autoclaving treatments show an increase in the observed contact angle in comparison to that of the pristine samples, with measured initial contact angles of 103.9˚±7.7˚, 105.4˚±6.2˚, and 96.0˚±15.2˚, respectively (S2 Fig). The treated samples' contact angle values remain consistent over time, whereas the pristine sample showed a marked decrease. No significant differences in the droplet volume over time are observed for the three samples. For all three samples, the inner surface rate of absorption was too rapid to allow for measurements by contact angle with the 20 μL droplet being absorbed during the first 1 ms measurement interval.

In summary, material characterization of the N95 respirator material revealed some helical length shortening of the respirator material and overall, no changes in neither the composition nor the crystallinity of any of the 3 layers of the respirator after dry heat treatment at 100˚C or after autoclaving. Additionally, the contact angle measurement of artificial saliva showed minimal changes in the behavior of liquid droplets on the respirator material under the same conditions. Taken together, it can be concluded that the filtration material itself was not much affected by either treatment. Material characterization was also performed on the Bacou Willson 801 N95 respirator (S4–S7 Figs) that showed similar results.

## Estimated filtration

Our data suggests that 100˚C dry heat treatment does not appreciably impact the respirator material and that autoclaving similarly does not appreciably impact the filter material. Applying the measured fit factors, we can estimate the hypothetical decrease in filtration efficiency assuming a perfect fit (i.e., perfect seal between face and mask). As outlined above, the quantitative fit test makes use of particles in the size range of ~40 nm that are commonly filtered with

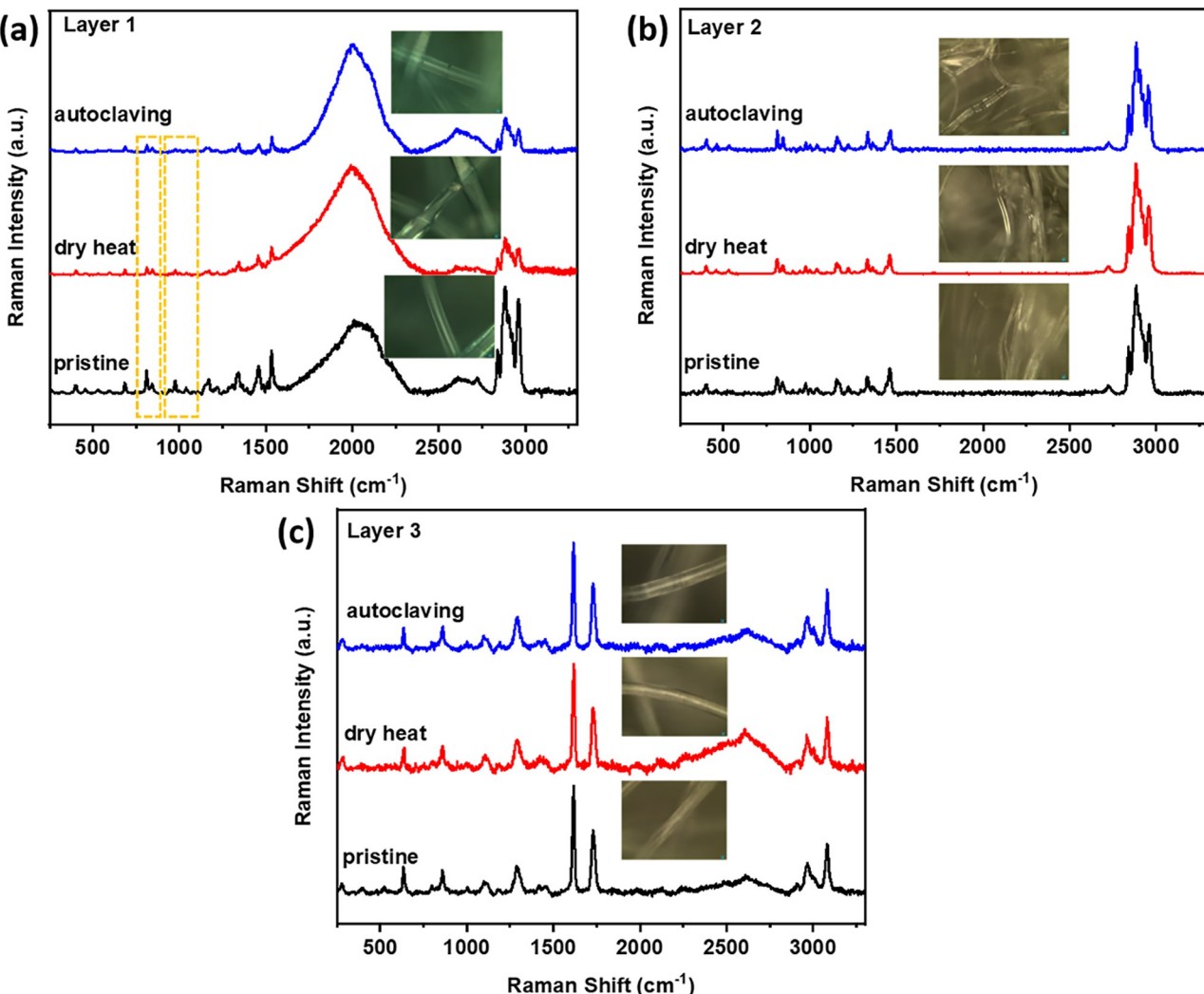

**Fig 4. Raman spectra of three layers in 3M 1860 N95 FFR material.** Spectra before dry heat, after dry heat, and autoclave heat treatment for the respective layers.

an efficiency of about 99.99% [23]. Estimated filtration efficiency was back-calculated from the definition of a fit factor, a ratio of the concentration of ambient particles to the concentration of particles inside of the respirator, thus is equal to 1–1/FitFactor (Table 2). Thus, a fit factor of 100 and 200 corresponds to a filtration efficiency of 99 and 99.5% respectively (both implying passing of the fit test). The estimated filtration efficiency for autoclaved respirators drops significantly below those thresholds, by 1.04% to 7.73%, while the estimated filter efficiencies dropped by <1%, if at all, in the dry heat-treated groups.

## Discussion

While most developed nations were able to deploy effective responses to mitigate supply chain shortages for PPE in the face of the COVID-19 pandemic, many parts of the world are still forced to reuse N95 respirators even after exposure to symptomatic patients [4,31]. PPE shortages, including limited supplies of N95-grade respirator masks, impacts a diverse set of healthcare facilities, from hospitals to nursing homes. The CDC acknowledges these shortages and

**Table 2. Estimated filter efficiency derived from fit factors obtained from the PortaCount Pro 8048.**

| Respirator | Condition | Estimated Filtration (%) | Δ Estimated Filtration (%) |
|---|---|---|---|
| 3M 1860 | Untreated | 99.50 | – |
| 3M 1870 | Untreated | 99.48 | – |
| BLS 120B | Untreated | 99.50 | – |
| Bacou Willson 801 | Untreated | 96.74 | – |
| 3M 1860 | Autoclaved | 91.67 | -7.83 |
| 3M 1870 | Autoclaved | 98.39 | -1.11 |
| BLS 120B | Autoclaved | 98.46 | -1.04 |
| Bacou Willson 801 | Autoclaved | 90.00 | -7.22 |
| 3M 1860 | 100°C | 99.50 | 0.00 |
| 3M 1870 | 100°C | 99.48 | -0.02 |
| BLS 120B | 100°C | 99.50 | 0.00 |
| Bacou Willson 801 | 100°C | 96.74 | -0.48 |
| 3M 1860 | 80°C | 99.50 | 0.00 |
| 3M 1870 | 80°C | 99.50 | 0.00 |
| BLS 120B | 80°C | 99.50 | 0.00 |
| Bacou Willson 801 | 80°C | 96.67 | -0.56 |

offers guidance, based on an institutions' burn rate (i.e., the rate at which N95 FFRs are used and disposed of) and crisis capacity strategies, whether N95 respirators are recommended for reuse and what methods for decontamination are authorized. The data in the current study offers an alternative and potentially more accessible method for decontamination of N95 respirators for reuse during crisis capacity. As the CDC recommends, limited FFR reuse should only be attempted when respirators are unsoiled, fit properly, and are undamaged (e.g., the straps and nosepiece are still intact and functional).

Consistent with previous studies, we observed a 2-log reduction of SARS-CoV-2 titer after treatment at 80°C for 60 minutes and undetectable virus following treatment at 100°C for 30 minutes, meeting the minimum previously suggested 5-log reduction in virus by the FDA [13–15]. Respirators subjected to dry heat maintained their gross structural integrity and the functionality of their straps and nosepieces after 4 cycles of sterilization. Similarly, dry heat sterilization after 4 cycles did not affect their fit as measured by quantitative fit testing. In contrast, autoclaved N95 respirators appeared to have some physical damages in the overall structure and shape of the respirators and failed quantitative fit testing in all respirator types tested. These results after autoclaving are consistent with other reports on most N95 respirator types that show degradation of the respirators [7,13,32].

Material characterization of 3M 1860 N95 respirators, performed by SEM, Raman spectroscopy, and XRD analysis, revealing some helical chain shortening, but no compositional or crystallite size changes in the microscopic structure of the 3 layers of the respirator after dry heat treatment. Similarly, autoclaving did not reveal any major changes in the material of the N95 respirator material. Contact angle characterization were also performed to evaluate any potential changes in the response to contact to liquids. The results demonstrated that the droplet volume remained consistent after either dry heat treatment or autoclave, although there was a slight increase in the contact angle after autoclave and dry heat treatment, suggesting that the absorption of liquids decreased after contact. Material characterization was also performed on the Bacou Willson 801 respirator (S4–S7 Figs) that revealed similar results.

Our data suggests that 100°C dry heat treatment does not significantly impact the fit or the material of the N95 respirators. Notably, the Bacou Willson 801 respirators failed quantitative

fit testing of a single subject under all conditions, despite being a NIOSH approved N95 FFR. These respirators may fit another user better, thus potentially yielding passing fit test scores. These results emphasize the importance to conduct individual fit tests after decontamination procedures as typically required in a health care setting. By comparison, all autoclaved respirators failed fit testing, despite having minimal changes in the material quality. Autoclaved respirators failed the respirator fit test and suggested that the change in filtration efficiency is up to 7.73%. However, the respirator material analysis does not corroborate significant changes in the filtration material that could lead to these decreases in filtration efficiency. This points to the fact, that most likely, autoclaving the respirators led to changes in the respirator fit by either altering the respirator mold, sealant, and/or straps. Taken together, these data suggest that autoclaving indeed has an impact on the fit of the respirator material. In combination with previous studies on the effects of autoclaving N95 respirators [13,32] and their deleterious impact on both fit and filtration, the results confirm that autoclaving is not a consistently viable method for the decontamination of N95 respirators for their reuse.

Although we assessed the function of decontaminated respirators by quantitative fit testing and material characterization, our study does not directly distinguish whether failed fit testing is due to the impairment of the filtration efficiency, including any impact on the electret properties of N95 respirators [16], or due to the failure of fit or some combination of both fit and filtration, although our data suggests that impact on fit as the most likely cause. In summary, dry heat sterilization is a potentially scalable, accessible, and effective method of decontaminating N95 respirators for up to 4 cycles in times of crisis and PPE shortages. Lastly, it is important to note that there is a possibility that through the process of convection, SARS-CoV-2 particles can be mobilized from surfaces and possible nosocomial infection can occur, especially without proper use of PPE [33].

## Supporting information

**S1 File. Fit factor assessment by application of a Scanning Mobility Particle Sizer Spectrometer (SMPS).**
(PDF)

**S1 Fig. XRD of three layers in 3M 1860 N95 FFR before and after dry heat and autoclaving.** Compared with the XRD patterns of the pristine masks (black), after dry air treatment (red) and autoclave/steam treatment (blue), the 3M 1860 N95 FFR materials showed no compositional changes and insignificant crystallite sizes changes after both types of thermal treatment. Specifically, the respective crystallite sizes of the pristine, dry air treated and steam treated layer are 15, 17 and 14 nm for layer 1 (**S1a Fig**); and 11, 11 and 8 nm for layer 2 (**S1b Fig**). This indicated that dry air treatment slightly increased the crystallize size at layer 1 with no significant change at layer 2. Interestingly, steam treatment decreased crystallite size for both layers 1 and 2 for the 3M 1860 N95 FFR materials. No crystallite size was calculated for layer 3 due to its more amorphous character with significant peak overlap, and no obvious change was observed on layer-3 between the pristine and heat-treated samples (**S1c Fig**).
(TIFF)

**S2 Fig. Contact angle measurements of 3M 1860 N95 FFR material.** In contact angle measurements for the 3M 1860 N95 FFR material, both the dry heat and steam treatments show an increase in the observed contact angle in comparison to that of the pristine, which evince an initial contact angle of 103.9°±7.7°, 105.4°±6.2°, and 96.0°±15.2°, respectively. The treated samples' contact angle values remain consistent over time, whereas the pristine sample showed a marked decrease. No significant difference is shown between the droplet volume over time

for the three samples. This observation suggests that the wettability of the pristine sample increases over time, but this behavior is ameliorated by the dry heat and steam treatments. For all three samples, the inner surface rate of absorption was too rapid to allow for measurements by contact angle with the 20 μl droplet being absorbed during the first 1000 μs measurement interval.
(TIFF)

**S3 Fig. Fit factor calculations derived from SMPS measurements of N95 respirators.** We derived the fit factor as defined in OSHA guidelines (see Methods section). Upper and lower bounds of the fit factor assumed the most conservative count estimates applying measured counts and their corresponding count error. Most conservative signifies, e.g., the greatest number of 40 nm particle in room air (including count uncertainty) over lowest number of 40 nm particles in respirator (subtracting count uncertainty).
(TIFF)

**S4 Fig. SEM characterizations of three layers in Bacou Willson 801 N95 FFR.** (**a-b**) SEM images of cross section. (**c-e**) SEM images of top down view of pristine Bacou Willson 801 N95 FFR. (**f-h**) SEM images of top down view of the of Bacou Willson 801 N95 after heat treatment at 100°C for 4 cycles. (**i-k**) SEM images of top down view of the of Bacou Willson 801 N95 after steam treatment. The morphologies of dry air heat treatment and steam (autoclave) treatment do not show obvious difference from SEM images, which indicates the morphologies are not changed under the dry air heat treatment and steam treatment methods used.
(TIFF)

**S5 Fig. XRD of three layers in Bacou Willson 801 N95 FFR material.** XRD of three layers in Bacou Willson 801 N95 (**a-c**) before and after dry heat and steam heat treatment. The XRD patterns of layer 1 and layer 2 indicated a semicrystalline character, and as marked therein, the major diffraction patterns were indexed to reflections from (110), (045), (130) and (-131) planes of the polypropylene phase (PDF #50–2397). Layer 1 showed larger crystallite size (16 nm) than layer 2 (4 nm), indicating layer 1 is more crystalline than layer 2 in the pristine mask (**S5a-b**, black curves). However, in layer-2 an extra peak at $2\theta = 20.07°$ was evident (**S5b**), corresponding to the (111) peak of polypropylene. The XRD patterns of the layer-3 also indicated a semicrystalline character, and as marked therein, the major diffraction patterns were indexed to reflections from the (010), (-110) and (100) planes of the polyester phase (PDF #50–2275) (**S5c**). Compared with the XRD patterns of pristine (untreated) Bacou Willson 801 N95 (black curve), the XRD patterns after dry air treatment (red) and steam treatment (blue) indicate higher crystallinity for layers 1 and 2 (**S5-b**). Specifically, the crystallite sizes of the respective pristine, dry heat treated, and steam treated 16, 18 and 19 nm for layer 1 (**S5a**) and 4, 8 and 12 nm for layer 2 (**S5a**). This indicated that steam treatment increases crystallite size more than dry air treatment at both layer 1 and layer 2, and the dry air treatment and steam treatment has more effect on crystallinity of layer 2 than layer 1. Layer 3 is more amorphous with significant peak overlap, so no crystallite size was calculated on layer 3, but an extra peak at $2\theta = 48.36$ were observed after dry air and steam treatment, which corresponds to the (200) peak of polyester. (**S5c**).
(TIFF)

**S6 Fig. Raman spectra of three layers in Bacou Willson 801 N95 FFR before and after dry air and steam heat (autoclave) treatment.** Based on the acquired Raman spectra, layer 1 and 2 of Bacou Willson 801 N95 FFR (**S6a & c**) have spectra features resembling polypropylene materials. After the dry air and steam treatment, the ratio of the two bands at 810 and 840 cm$^{-1}$ layer 1 decreased, along with the decreasing intensity of the 972 cm$^{-1}$ peak (**S6a**), as highlighted

in the yellow dashed regions, suggesting a shorting of the helical chain conformation of polypropylene after heat treatment [28]. The regularity bands at 973, 998, 841, and 1220 $cm^{-1}$ were previously assigned to the helical chains of 5, 10, 12, 14 monomeric units of polypropylene, respectively. The 2nd layer of the Bacou Willson 801 N95 FFR material (**S6**) showed broader and weaker peaks than those in layer 1, which potentially suggested lower crystallinity in this layer consistent with the narrower thickness of the fibers [34]. No significant changes were noted after the heat treatment in this layer. The 3rd layer of the Bacou Willson 801 N95 FFR can be assigned to polyester (**S6c**),[35] as indicated by the strong C = C stretching band (ring deformation) at 1615 $cm^{-1}$ and C = O stretching band at 1730 $cm^{-1}$. Similarly, no significant differences in Raman spectra were observed in the bulk structure of layer 3 before and after heat treatment, suggesting minimal changes in crystallinity and bond orientation implying that layers 2 and 3 of Bacou Willson 801 N95 FFR are stable under heat treatment. (TIFF)

**S7 Fig. Contact angle measurements of Bacou Willson 801 N95 FFR material.** Contact angle and droplet volume over time of outer (**a & b**) and inner (**c & d**) surfaces of Bacou Willson 801 N95 FFR before and after dry heat and steam treatment. There is not an observed significant difference between the contact angle of the pristine, dry heat treated, and steam treated Bacou Willson 801 N95 FFR samples, which evince an initial contact angle of 121.4˚ ±12.6˚, 121.2˚±9.5˚, and 113.1˚±8.3˚, respectively, and remain constant over time. Though the steam treated (autoclaved) sample shows the greatest reduction in contact angle, suggesting an increase in surface adsorption of the artificial saliva, this value still lies within the error of the pristine (untreated) measurement. Furthermore, the same similarity in surface absorption is shown by the observation of the volume of the liquid droplet over time. Greater variability is observed when measurements are taken of the inner surfaces of the Bacou Willson 801 N95 FFR samples, (**S7c & d**). Whilst there are no significant initial differences between the pristine and heat-treated samples, 106.9˚±6.8˚ and 111.9˚±15.1˚, respectively, both samples show a decreasing trend over time. The most rapid decrease is observed within the pristine sample, which achieves a value of 64.4˚±15.2˚ after 1 minute. This increase in surface adsorption and wettability of the inner surfaces is further supported by the increase rate of surface absorption suggest by negative slope of the volume over time figure. The steam treated sample rate of surface absorption was too rapid to allow for measurements by contact angle with the 20 μl droplet being absorbed with first 1000 μs measurement interval. The wetting properties of the inner mask to outer mask surfaces suggest the inner surfaces draw respiratory expulsions away from the user whereas the outer surfaces can repel respiratory expulsions toward the user from other sources. Dry heat treatments decrease the absorption of the inner layer; however, steam treatments induce a distinct increase of the absorption of the inner layer. (TIFF)

## Acknowledgments

We especially appreciate Ellen O'Hare and Kathy Terwilliger for assisting with the quantitative fit testing. We appreciate Emily Costa and Josephine Aller for assisting with SMPS measurements. We would also like to acknowledge Tenny Environmental, New Columbia, PA for the use of the use of a T2 series small dry-heat sterilizer for disinfection studies.

## Author Contributions

**Conceptualization:** John G. Yuen, Amy C. Marshilok, Jeronimo Cello, Kenneth J. Takeuchi, Daniel A. Knopf, Kenneth R. Shroyer.

**Data curation:** John G. Yuen.

**Formal analysis:** John G. Yuen, Peter Todd Benziger, Shan Yan, Chavis A. Stackhouse, Kim Kisslinger, David C. Bock, Esther S. Takeuchi, Kenneth J. Takeuchi, Lei Wang.

**Funding acquisition:** Amy C. Marshilok.

**Investigation:** John G. Yuen, Peter Todd Benziger, Shan Yan, Chavis A. Stackhouse, Kim Kisslinger, David C. Bock, Sruthi Babu, Daniel A. Knopf.

**Methodology:** John G. Yuen, Jeronimo Cello, Esther S. Takeuchi, Glen Itzkowitz, Daniel A. Knopf, Kenneth R. Shroyer.

**Project administration:** Amy C. Marshilok, Kenneth R. Shroyer.

**Resources:** Amy C. Marshilok.

**Supervision:** Esther S. Takeuchi, Kenneth J. Takeuchi, David Thanassi, Daniel A. Knopf, Kenneth R. Shroyer.

**Writing – original draft:** John G. Yuen, Peter Todd Benziger, Daniel A. Knopf, Kenneth R. Shroyer.

**Writing – review & editing:** John G. Yuen, Amy C. Marshilok, Peter Todd Benziger, Shan Yan, Jeronimo Cello, Glen Itzkowitz, David Thanassi, Daniel A. Knopf, Kenneth R. Shroyer.

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
