## [Decision Letter · Decision Letter 0]

29 Oct 2021

PONE-D-21-27839Dry heat sterilization as a method to recycle N95 respirator masks: the importance of fitPLOS ONE

Dear Dr. Shroyer,

Thank you for submitting your manuscript to PLOS ONE. After careful consideration, we feel that it has merit but does not fully meet PLOS ONE’s publication criteria as it currently stands. Therefore, we invite you to submit a revised version of the manuscript that addresses the points raised during the review process.

We look forward to receiving your revised manuscript.

Kind regards,

Swatantra Pratap Singh, Ph.D.

Academic Editor

PLOS ONE

Journal Requirements:

Additional Editor Comments (if provided):

Manuscript has been written well and can be accepted after miner revision as per reviewers comments.

Reviewers' comments:

Reviewer's Responses to Questions

**Comments to the Author**

1. Is the manuscript technically sound, and do the data support the conclusions?

Reviewer #1: Yes

Reviewer #2: Yes

2. Has the statistical analysis been performed appropriately and rigorously? 

Reviewer #1: Yes

Reviewer #2: No

3. Have the authors made all data underlying the findings in their manuscript fully available?

Reviewer #1: No

Reviewer #2: Yes

4. Is the manuscript presented in an intelligible fashion and written in standard English?

Reviewer #1: Yes

Reviewer #2: Yes

5. Review Comments to the Author

Reviewer #1: The manuscript is technically sound and well written . However, in the results section the complete data for thermal stability on N95 Respirators seems to be missing. Thus same can be provided in the supporting information.

Apart from this , following are some of the queries which may improve the paper further.

Comment 1: Introduction section line 83; “…OSHA compliant fit factors that quantitatively evaluate whether a respirator fits properly…”. OSHA is used for the first time Thus it is advised to write the full form of OSHA over here.

Comment 2: Materials and Methods section line 123; “VeroE6 cells were routinely cultured in DMEM containing Glutagro (Corning) and 8%...”. writing the full form of DMEM media is recommended.

Comment 3: N95 FFR Quantitative Fit Tests line 157-158; “Quantitative fit testing was performed on the same operator for all mask types and for all conditions:

dry heat treated (n = 3), untreated (n = 1), and autoclaved (n = 1) respirators”. As comparison is done between dry heat treated, untreated, and autoclaved respirators taking the value of n 3, 1, and 1 respectively, then why the heterogeneity is there in terms of the sample size n?

Comment 4: N95 FFR Quantitative Fit Tests line 159-165; “Quantitative fit testing procedures for N95 respirators were performed according to Occupational Safety…. passes OSHA guidelines”. It would be preferable for the readers if the authors could add the reference in this paragraph.

Comment 5: N95 FFR Quantitative Fit Tests line 166. Please check the formula of the Overall Fit factor and correct the running variable (or summation index). Also, it is advised to the author to describe the term n in the text as this will help the reader to understand better.

Comment 6: N95 FFR Quantitative Fit Tests line 168. It would be preferable for the readers if the authors could add the reference for Table 1.

Comment 7: Estimated Filtration line 285-286. “A fit factor of 100 and 200 corresponds to a filtration efficiency of 99 and 99.5% respectively (both imply passing of the fit test)”. How this relation between fit factor and filtration efficiency is drawn? It would be preferable for the readers if the authors could offer additional detail and explain the reason behind this trend.

Comment 8: Estimated Filtration line 290. In table 2 there is no indication of the calculation used to determine estimated filtration (%). Thus, it would be preferable for the readers if the authors could include the information of calculation done.

Comment 9: Please check the uniformity of the References. For instance: Few details of reference 11 are missing.

Reviewer #2: The work submitted is interesting and can be accepted with minor revisions. Please check the notations throughout the manuscript and keep it uniform.

Line 71: Please mention the time as it is a critical parameter for autoclaving

Follow the units in the same format throughout

Please provide a reference for Table 1 with a source

Please improve figure one

6. PLOS authors have the option to publish the peer review history of their article (what does this mean?). If published, this will include your full peer review and any attached files.

Reviewer #1: No

Reviewer #2: No

---

## [Author Response · Author response to Decision Letter 0]

24 Nov 2021

Manuscript PONE-D-21-27839

Response to Reviewers 

Dear Dr. Singh and Reviewers, 

Thank you for the feedback and for giving us the opportunity to submit a revised draft of the manuscript “Dry heat sterilization as a method to recycle N95 respirator masks: the importance of fit” for publication in PLOS ONE. We appreciate the time and effort that you and the reviewers dedicated to the review of the manuscript, and we are grateful for the insightful comments. We have incorporated the suggestions made by the reviewers and have outlined them below and within the manuscript. 

Reviewer #1: 

The manuscript is technically sound and well written. However, in the results section the complete data for thermal stability on N95 Respirators seems to be missing. Thus same can be provided in the supporting information.

Author response: Thank you very much! As for the data for thermal stability, the experiments were done in triplicate, where SARS-CoV-2 virus was recovered by cutting each virus-treated spot, from each N95 mask and submerging the samples in media. Briefly, samples were incubated for 5 minutes at room temperature then rocked gently to recover virus. Plaque assays were performed to quantify the amount of virus recovered by performing serial dilutions of recovered virus and infecting VeroE6 cells for one hour. To quantify the amount of virus recovered, the overlay was removed, and the plaques were visualized by staining VeroE6 cells with 0.5% crystal violet and 0.8% glutaraldehyde in 50% methanol. Total PFU/mL for each condition was calculated by averaging the mean PFU/mL recovered for each biological replicate. Additional details, including the description above, are written in the Methods section “SARS-CoV-2 Thermal Stability/Viral Recovery and Quantification.”

Comment 1: Introduction section line 83; “…OSHA compliant fit factors that quantitatively evaluate whether a respirator fits properly…”. OSHA is used for the first time Thus it is advised to write the full form of OSHA over here.

Author response: Thank you for pointing this out, we have written out “Occupational Safety and Health Administration” as advised in line 84 of the revised manuscript. 

Comment 2: Materials and Methods section line 123; “VeroE6 cells were routinely cultured in DMEM containing Glutagro (Corning) and 8%...”. writing the full form of DMEM media is recommended.

Author response: Thank you for pointing this out, we have written out “Dulbecco’s Modified Eagle Medium” as advised in line 140 of the revised manuscript.

Comment 3: N95 FFR Quantitative Fit Tests line 157-158; “Quantitative fit testing was performed on the same operator for all mask types and for all conditions:

dry heat treated (n = 3), untreated (n = 1), and autoclaved (n = 1) respirators”. As comparison is done between dry heat treated, untreated, and autoclaved respirators taking the value of n 3, 1, and 1 respectively, then why the heterogeneity is there in terms of the sample size n?

Author response: Thank you for pointing this out. At the time the experiment was conducted, there was a shortage of N95 respirators, and the supply of respirators were tightly regulated by Environmental Health and Safety at our institution. We therefore decided to take a conservative approach, as we must destroy the N95 respirators in order to perform fit testing by puncturing the masks in order to affix a valve onto the respirator. Similarly, autoclaving a N95 respirator will undoubtedly damage the respirator as reported in the literature (doi: 10.1016/j.jhin.2020.06.030, 10.1371/journal.pone.0243965) and as observed in our experiment by the deformation of the respirator.

Comment 4: N95 FFR Quantitative Fit Tests line 159-165; “Quantitative fit testing procedures for N95 respirators were performed according to Occupational Safety…. passes OSHA guidelines”. It would be preferable for the readers if the authors could add the reference in this paragraph.

Author response: Thank you for pointing this out. We have added the appropriate citation. 

Comment 5: N95 FFR Quantitative Fit Tests line 166. Please check the formula of the Overall Fit factor and correct the running variable (or summation index). Also, it is advised to the author to describe the term n in the text as this will help the reader to understand better.

Author response: Thank you for pointing this out. We have clarified the definition of n in the text as the number of exercises and corrected the typo in the summation index. 

Comment 6: N95 FFR Quantitative Fit Tests line 168. It would be preferable for the readers if the authors could add the reference for Table 1.

Author response: Thank you for pointing this out. We have added the citation to Table 1. 

Comment 7: Estimated Filtration line 285-286. “A fit factor of 100 and 200 corresponds to a filtration efficiency of 99 and 99.5% respectively (both imply passing of the fit test)”. How this relation between fit factor and filtration efficiency is drawn? It would be preferable for the readers if the authors could offer additional detail and explain the reason behind this trend.

Author response: Thank you for pointing this out. We have clarified that the estimated filtration can be calculated based off the definition of a fit factor (the ratio of the concentration of ambient particles to the concentration of particles inside the respirator), thus is equal to 1 – 1/FitFactor. These changes are reflected in lines 349-351.

Comment 8: Estimated Filtration line 290. In table 2 there is no indication of the calculation used to determine estimated filtration (%). Thus, it would be preferable for the readers if the authors could include the information of calculation done.

Author response: We think this is an excellent suggestion and have addressed it with the same clarifications we made in response to Comment 7.

Comment 9: Please check the uniformity of the References. For instance: Few details of reference 11 are missing.

Author response: Thank you for pointing this out. We have double checked the references and added the missing details to references 11 and 2. 

Reviewer #2: The work submitted is interesting and can be accepted with minor revisions. 

Author response: Thank you very much!

Please check the notations throughout the manuscript and keep it uniform.

Author response: Thank you for pointing this out. We have made some revisions on the notation and have clarified the usage of mask, respirator, and FFR according to their most precise and accurate definitions as most commonly used by the CDC, NIOSH, OSHA and the manufacturers. 

Line 71: Please mention the time as it is a critical parameter for autoclaving

Author response: Thank you for pointing this out. The parameters are detailed more thoroughly in the Methods section, where we have stated that the autoclave time is for 30 minutes. The mention of autoclave conditions in line 71 are to point out in the Introduction that autoclaving involves both steam and pressure and typical conditions are given. 

Follow the units in the same format throughout

Author response: Thank you for pointing this out, we have double checked the units and have made revisions in the text. 

Please provide a reference for Table 1 with a source

Author response: Thank you for pointing this out. We have added the appropriate citation. 

Please improve figure one

Author response: Thank you for the suggestion. We have decided that this figure does not add much to the manuscript, and we have therefore removed it.

---

## [Decision Letter · Decision Letter 1]

14 Dec 2021

Dry heat sterilization as a method to recycle N95 respirator masks: the importance of fit

PONE-D-21-27839R1

Dear Dr. Shroyer,

We’re pleased to inform you that your manuscript has been judged scientifically suitable for publication and will be formally accepted for publication once it meets all outstanding technical requirements.

Kind regards,

Swatantra Pratap Singh, Ph.D.

Academic Editor

PLOS ONE

Reviewers' comments:

Reviewer's Responses to Questions

**Comments to the Author**

1. If the authors have adequately addressed your comments raised in a previous round of review and you feel that this manuscript is now acceptable for publication, you may indicate that here to bypass the “Comments to the Author” section, enter your conflict of interest statement in the “Confidential to Editor” section, and submit your "Accept" recommendation.

Reviewer #1: All comments have been addressed

2. Is the manuscript technically sound, and do the data support the conclusions?

Reviewer #1: Yes

3. Has the statistical analysis been performed appropriately and rigorously? 

Reviewer #1: Yes

4. Have the authors made all data underlying the findings in their manuscript fully available?

Reviewer #1: Yes

5. Is the manuscript presented in an intelligible fashion and written in standard English?

Reviewer #1: Yes

6. Review Comments to the Author

Reviewer #1: All the comments has been addressed well. However, the suggestion regarding the the complete data for thermal stability on N95 Respirators could have been addressed in a more effective way.

7. PLOS authors have the option to publish the peer review history of their article (what does this mean?). If published, this will include your full peer review and any attached files.

Reviewer #1: No

---

## [Editor Report · Acceptance letter]

16 Dec 2021

PONE-D-21-27839R1 

Dry heat sterilization as a method to recycle N95 respirator masks: the importance of fit 

Dear Dr. Shroyer:

I'm pleased to inform you that your manuscript has been deemed suitable for publication in PLOS ONE. Congratulations! Your manuscript is now with our production department. 

Kind regards, 

on behalf of

Dr. Swatantra Pratap Singh 

Academic Editor

PLOS ONE